# Kinase Inhibitor Treatment of Patients with Advanced Cancer Results in High Tumor Drug Concentrations and in Specific Alterations of the Tumor Phosphoproteome

**DOI:** 10.3390/cancers12020330

**Published:** 2020-02-01

**Authors:** Mariette Labots, Thang V. Pham, Richard J. Honeywell, Jaco C. Knol, Robin Beekhof, Richard de Goeij-de Haas, Henk Dekker, Maarten Neerincx, Sander R. Piersma, Johannes C. van der Mijn, Donald L. van der Peet, Martijn R. Meijerink, Godefridus J. Peters, Nicole C.T. van Grieken, Connie R. Jiménez, Henk M.W. Verheul

**Affiliations:** 1Department of Medical Oncology, Cancer Center Amsterdam, Amsterdam UMC, Vrije Universiteit Amsterdam, De Boelelaan 1117, 1081 HV Amsterdam, The Netherlands; m.labots@amsterdamumc.nl (M.L.); t.pham@amsterdamumc.nl (T.V.P.); r.honeywell@amsterdamumc.nl (R.J.H.); j.knol@amsterdamumc.nl (J.C.K.); r.beekhof@amsterdamumc.nl (R.B.); r.dehaas@amsterdamumc.nl (R.d.G.-d.H.); henk.dekker@amsterdamumc.nl (H.D.); mneerincx@spaarnegasthuis.nl (M.N.); s.piersma@amsterdamumc.nl (S.R.P.); j.vandermijn@amsterdamumc.nl (J.C.v.d.M.); gj.peters@amsterdamumc.nl (G.J.P.); 2Department of Surgery, Cancer Center Amsterdam, Amsterdam UMC, Vrije Universiteit Amsterdam, De Boelelaan 1117, 1081 HV Amsterdam, The Netherlands; dl.vanderpeet@amsterdamumc.nl; 3Department of Radiology, Cancer Center Amsterdam, Amsterdam UMC, Vrije Universiteit Amsterdam, De Boelelaan 1117, 1081 HV Amsterdam, The Netherlands; mr.meijerink@amsterdamumc.nl; 4Department of Pathology, Cancer Center Amsterdam, Amsterdam UMC, Vrije Universiteit Amsterdam, De Boelelaan 1117, 1081 HV Amsterdam, The Netherlands; NCT.vanGrieken@amsterdamumc.nl; 5Department of Medical Oncology, RadboudUMC, Radboud University, Geert Grooteplein Zuid 8, 6525 GA Nijmegen, The Netherlands

**Keywords:** mass spectrometry, protein kinase inhibitor, phosphoproteomics, tumor drug concentration, cancer, tyrosine phosphorylation

## Abstract

Identification of predictive biomarkers for targeted therapies requires information on drug exposure at the target site as well as its effect on the signaling context of a tumor. To obtain more insight in the clinical mechanism of action of protein kinase inhibitors (PKIs), we studied tumor drug concentrations of protein kinase inhibitors (PKIs) and their effect on the tyrosine-(pTyr)-phosphoproteome in patients with advanced cancer. Tumor biopsies were obtained from 31 patients with advanced cancer before and after 2 weeks of treatment with sorafenib (SOR), erlotinib (ERL), dasatinib (DAS), vemurafenib (VEM), sunitinib (SUN) or everolimus (EVE). Tumor concentrations were determined by LC-MS/MS. pTyr-phosphoproteomics was performed by pTyr-immunoprecipitation followed by LC-MS/MS. Median tumor concentrations were 2–10 µM for SOR, ERL, DAS, SUN, EVE and >1 mM for VEM. These were 2–178 × higher than median plasma concentrations. Unsupervised hierarchical clustering of pTyr-phosphopeptide intensities revealed patient-specific clustering of pre- and on-treatment profiles. Drug-specific alterations of peptide phosphorylation was demonstrated by marginal overlap of robustly up- and downregulated phosphopeptides. These findings demonstrate that tumor drug concentrations are higher than anticipated and result in drug specific alterations of the phosphoproteome. Further development of phosphoproteomics-based personalized medicine is warranted.

## 1. Introduction

With the identification of driver DNA alterations and molecular subtypes that predict benefit from PKIs, great strides towards realization of personalized or precision medicine have been made [1]. Focus in precision medicine has shifted from identification of single genomic events to a more comprehensive assessment of tumor biology through integration of genomic, transcriptomic and proteomic data [2,3,4]. However, accurate predictive biomarkers are warranted to guide targeted therapy selection for further improvement of treatment outcome as well as for prevention of unnecessary toxicity. Given the fact that PKIs are promiscuous compounds, whose off-target effects can be beneficial and harmful at the same time, one of the main optimization challenges is how to evaluate their activity in patients [5]. In addition, predictive biomarker research efforts often focus on molecular mechanisms of sensitivity and resistance, while tumor drug concentration receives minor attention, despite consequences for its efficacy [6]. With regard to PKIs, pre- and on-treatment tumor biopsies may provide the opportunity to analyze drug concentration as well as inhibition of (off-) target kinases in target tissue. Actual achievement of (target) kinase inhibition is dependent on multiple variables, including oral bioavailability and tumor cell exposure of the drug, as well as drug specificity for and abundance of active kinases in the tumor [7]. Identification of predictive biomarkers for PKI treatment will require an approach that can take these factors into consideration. Liquid chromatography coupled to tandem mass spectrometry (LC-MS/MS) has been the reference method for determination of drug concentrations in patient samples [8]. Mass spectrometry (MS)-based phosphoproteomics has more recently emerged as an approach of molecular tumor profiling to obtain insight in aberrantly activated signaling pathways and potential drug targets. This is achieved through global analysis of phosphorylated proteins that form the basis for cellular signaling activities and protein-protein interaction [9]. In particular, phosphotyrosine-(pTyr)-phospho-proteomics provides an opportunity for the identification of patient subgroups likely to benefit from tyrosine kinase inhibitors [10]. The potential of this high-throughput method has first been evidenced by the identification of the Anaplastic lymphoma kinase (ALK), Reactive oxygen species (ROS) and Platelet derived growth factor alpha (PDGFRα) mediated Non-small cell lung cancer (NSCLC) subtypes in 2007 [11] and just recently by identification of 6 kinases prognostic for the outcome of triple negative breast cancer [12]. We previously demonstrated that MS-based profiling of the tyrosine phosphoproteome in tumor biopsies is feasible and provides patient-specific profiles, enabling its further development for treatment selection purposes [13]. In the preclinical setting, for example using chemical proteomics in triple negative breast cancer cell lines [14], alterations of the kinome in response to PKI inhibition have been shown. However, there are no data available on the effects of such treatment on the tyrosine phosphoproteome of patient tumors. In the present study, we use LC-MS/MS to analyze tumor concentrations and to evaluate the effect of the registered PKIs sorafenib (SOR), erlotinib (ERL), dasatinib (DAS), vemurafenib (VEM), sunitinib (SUN) and everolimus (EVE) on pTyr-phosphoproteomic profiles of tumor biopsies in patients with advanced cancer in order to get a better understanding of their mechanism of action in the clinical setting. 

## 2. Results 

### 2.1. Patient Accrual and Characteristics

In this clinical trial, patients underwent a tumor needle biopsy before and after 10–14 days of study treatment with 1 of the selected PKIs (Figure 1). Thirty-six patients with advanced cancer started PKI treatment after pre-treatment biopsy of 41 patients whom signed informed consent. Five of 41 patients did not start study treatment due to the impossibility of a pre-treatment tumor biopsy based on the judgement of the interventional radiologist. Six cohorts were completed, i.e., with 5 evaluable patients defined as successfully performed pre- and on-treatment tumor biopsy. Tumor re-biopsy was performed in 86% (31/36) of patients (Appendix A). Characteristics of patients for whom on-treatment tumor biopsies were obtained and whom were evaluable for the primary endpoint are summarized in Table 1. No adverse events CTC grade 3–4 resulted from the biopsies or from study treatment. 

### 2.2. Accumulation of Protein Kinase Inhibitors in Tumor Tissue

In 28 of 31 patients (90%) whom underwent both pre- and on treatment biopsy, tumor PKI concentrations could be determined by LC-MS/MS after 10–14 days of standard daily oral PKI administration. Median tumor concentration ranged between 2.0–10.0 µM for the SOR, ERL, DAS, SUN and EVE cohorts, while for VEM this was 1.3 mM. These were > 10-fold (range 2.1–177.6 ×) higher than median plasma concentrations (Table 2). Eighty-seven percent of re-biopsied patients (27/31) also consented for skin biopsy. Tumor and skin concentrations were largely comparable with a median 1.6-fold (range: 1.5–5.8 ×) higher accumulation in tumor tissue for most cohorts. PKI concentration data for individual patients are shown in Appendix A. These plasma and serum concentrations were within the range found in literature [15,16,17,18,19,20], while no robust data are available in the literature for tumor and skin concentrations of these drugs during treatment. The fact that these tumor concentrations exceed concentrations that are required for target kinase inhibition, indicates that—due to the promiscuity of PKIs at these higher concentrations [5,21]—far more than the supposed target kinases will be affected upon treatment with these agents.

### 2.3. Tyrosine-Phosphoproteomic Profiling of Pre- and On-Treatment Tumor Biopsies

To obtain insight in the real-life PKI activity, at the tumor level in patients, we further analyzed the pre- and on-treatment biopsies by MS-based phosphotyrosine (pTyr) phosphoproteomics using a miniaturized workflow for samples with limited input material. pTyr-phosphoproteomic analysis was successful for all patients in 5 out of 6 aforementioned cohorts; results of only 2 EVE-treated patients (1 pair, 1 single biopsy) could be retrieved. This cohort was therefore left out of further phosphoproteomics analyses. In the five treatment cohorts, pTyr-phosphoproteomic profiling was performed with a median paired protein input of 2 mg per biopsy (range 0.6–2.9) for ERL, SOR and DAS and 1 mg (0.4–2.0) for VEM and SUN. Per biopsy, on average 467 ± 125 phoshopeptides were identified in ERL/SOR/DAS and 206 ± 116 in VEM/SUN, respectively. The difference in number of identified peptides reflects the positive relation between protein input and phosphopeptide identification (Appendix A). In this dataset, a total of 37,806 peptides were identified, of which 2326 were phosphorylated (6%). Identified and quantified phosphopeptides and – sites can be found in Appendix A. 

### 2.4. Drug-Specific Alterations of Peptide Phosphorylation upon Treatment

Analyzing the tyrosine phosphoproteome of all tumor biopsies, unsupervised clustering of the phosphopeptide intensities showed that within the drug cohorts, paired pre- and on-treatment tumor biopsies from individual patients clustered, indicating that PKI-induced changes to the phosphoproteome were smaller than inter-patient differences (Figure 2A). Despite the seemingly similar pre- and on-treatment profiles of individual patients, PKI treatment resulted in up- or downregulation of tens to hundreds of phosphopeptides per patient with a fold-change of on-treatment/pre-treatment intensity (Fc) of > 1.5–5 upon treatment (data not shown). The identification of phosphopeptides that were differentially regulated due to specific PKI treatment was shown by a stringent selection of (up- or down-) regulated phosphopeptides with a Fc of > 5 in at least 3 of 5 patients (≥60%) per cohort. Supervised clustering analysis of these highly regulated phopshopeptides separated pre- and on-treatment groups in all drug cohorts (Figure 2B; Appendix A). Figure 3 shows the marginal overlap between the SOR, ERL, DAS and VEM treatment groups of Fc > 1.5 up- or downregulated phosphopeptides in ≥ 3 of 5 patients within each cohort, confirming that, even at a much less stringent selection, these 39–74 up- and 4–135 downregulated phosphopeptides were drug-specific. These data support preclinical evidence that at the concentrations we have measured (Table 2), these drugs do affect multiple kinases [5,22]. Analysis of Pearson’s correlations between reference samples measured along with the drug cohorts on subsequent days indicated that the differences in regulated peptides between the drug cohorts could not be attributed to day to day variation or batch effects (Appendix A). 

### 2.5. Protein Networks of Up- and Downregulated Phosphopeptides

To allow more insight in the biological interpretation of phosphopeptide regulations, peptides were mapped per cohort to proteins and visualized as protein interaction networks (Appendix A), wherein the nodes represent > 1.5 Fc up – and downregulated peptides identified in ≥ 3 patients. These analyses are hypothesis generating and need further confirmation in follow-up experiments. For example, in the ERL cohort, downregulated proteins in 3 of 5 patients (3/5) included epidermal growth factor receptor (EGFR) and Signal transducer and activator of transcription (STAT) 3 as well as the highly connected node MET (5/5). Upregulated proteins included the non-receptor tyrosine kinase protein, Gardner-Rasheed feline sarcoma (FGR) in 4/5 while VIM was upregulated in 5/5 patients. In the SOR network downregulation of Kinase insert domain receptor (KDR), also known as vascular endothelial growth factor receptor-2 (VEGFR-2) (3/5), and Mitogen-activated protein kinase-13 (MAPK13) and Janus kinase-2 (JAK2) in 4/5 were detected. Upregulated peptides included AXL (3/5), STAT6 (4/5) as well as Protein phosphatase 1 catalytic subunit alpha (PPP1CA) and Neural precursor cell expressed developmentally down-regulated protein 9 NEDD9 (5/5). Analysis of the downregulated peptides in the DAS-cohort revealed protein tyrosine kinase 2 (PTK2), also known as focal adhesion kinase (FAK) as one of the highest-connected nodes in this network, regulated in 4/5. Downregulation of the proto-oncogene tyrosine-protein kinase (Src) -family kinases LYN and FGR was observed in 3/5, respectively, while discoidin domain receptor1 (DDR1), a receptor tyrosine kinase previously shown also to be inhibited by dasatinib [23], was downregulated in 5/5 patients. Upregulated proteins in this cohort included EGFR, which was a highly connected node and regulated in 5/5, in addition to Phosphatidylinositol 3-kinase regulatory subunit gamma (PIK3R3) (4/5) and the member of the EGFR-family, receptor tyrosine-protein kinase erbB-3 (ERBB3) (3/5). 

### 2.6. Correlation between Tumor Concentration and Inhibition of Peptide Phosphorylation

The total number of up- or downregulated phosphopeptides per patient did not correlate with individual tumor concentrations (data not shown). However, several phosphopeptides showed reduced intensity in on- vs pre-treatment samples with increasing tumor concentration (Appendix A). For example, 14 phosphopeptides demonstrated such ‘anti-correlation’ in the ERL cohort, including, in 3/5 patients, peptides related to EGFR, PTK2, STAT 3 and, in 4/5 patients, Protein kinase C delta type (PRKCD). Similarly, *up*regulated peptides correlating with increased tumor concentration may provide information on stress or potential resistance mechanisms induced by drug treatment. In SOR, eight such ‘correlating’ peptides included NEDD9, a scaffold protein suggested to have a coordinating role in several cellular signaling processes [24], HIST1H4A and PPP1CA, a serine/threonine specific protein phosphatase, and all three were upregulated in 5/5 patients.

## 3. Discussion

This study not only elaborates on previous efforts in the quest for predictive biomarkers to optimize treatment with protein kinase inhibitors (PKIs), but also represents patients’ willingness to altruistically participate in research, i.e., without (immediate) benefit or impact for treatment, consistent with previous findings [25,26]. As one of its most important findings, this study contributes to the paucity of data on the penetration of four tyrosine (erlotinib, sorafenib, dasatinib, sunitinib) and two serine/threonine (vemurafenib, everolimus) kinase inhibitors in tumor tissue (for target profile see Appendix A). At plasma and serum concentrations that were within the range found in literature [15,16,17,18,19,20], median tumor concentrations were between 2–10 µM for the ERL, SOR, DAS, SUN and EVE cohorts; for VEM this exceeded 1 mM. Tumor erlotinib and sunitinib concentrations were consistent with small sample-sized studies suggesting 4–20× higher concentrations of (mostly EGFR) tyrosine kinase inhibitors in tumor than in plasma [27,28,29]. Since plasma concentrations do not predict tumor concentrations accurately, our results question the relevance of therapeutic drug monitoring (TDM) using plasma to optimize clinical activity of PKIs [30]. Furthermore, these results reveal that tumor drug concentrations well exceed concentrations required for inhibition of the assumed drug specific kinase targets [27,28]. Due to the promiscuity of PKIs at these higher concentrations [5,21], their biological activity will also be due to inhibition of other kinases for which these TKIs have lower affinity [31,32,33]. 

In this large-scale MS-based phosphoproteomic analysis of pre- and on treatment tumor needle biopsies obtained in a clinical trial, we studied to what extent these high tumor drug concentrations indeed may alter activity of many more kinases than their known targets. Regulation of multiple kinases was demonstrated by pTyr-phosphoproteomics performed in pre- and on-treatment samples of 31 patients, revealing up- and downregulation in tens to hundreds of phosphopeptides dependent on the drug that was administered. Thus far, one study has reported pre- and on-treatment *global* phosphoproteomics in a single patient with advanced hepatocellular carcinoma treated with sorafenib [34]. Furthermore, we show the potential of this approach to identify patient-specific profiles. Treatment-induced differential regulation of the phosphoproteome could only in part be related to known targets (tyrosine kinases) of administered drugs, but were in addition found to consist of peptides related to several other (less sensitive target) kinases or substrates. The latter also fits the concept that promiscuous rather than targeted drugs are most effective for (durable) treatment of cancer [35]. To what extent these changes in kinase activity, as shown by the alterations in peptide phosphorylation, are directly due to drug-kinase interaction or indirectly caused through a sequence of effects on the downstream signaling cascade, is difficult to determine. However, based on their tumor concentrations and differential activity, one can conclude that at least part of these changes are due to direct interactions. Regulation of multiple substrates, either direct or indirect, was observed in all cohorts, including for erlotinib, which has also been shown to bind to several other kinases with similar or higher affinity than EGFR [31,36]. The observation of multiple inhibitory effects is considered beneficial in terms of potential anticancer activity, but the observed upregulation of phosphorylated peptides may reflect stimulatory and pro-survival events, as for example visible in the networks of the SOR and VEM cohorts (Appendix A). PKI-induced upregulation of phosphorylated proteins has been reported in in vitro experiments by Zhang et al and by ourselves [22,37]. These potential pro-survival signals support the search for alternative combination treatment strategies in order to circumvent these potential resistant mechanisms [38]. In addition, these findings may be taken into consideration for novel kinase inhibitor drug development to prevent these type of unwanted drug-effects.

Integration of both on-treatment sample-based analyses did not reveal a direct correlation between total tumor PKI concentration and total number of up- or downregulated peptides. This may in part be explained by our observation that (even) the lowest drug concentrations measured in the obtained biopsies are amply sufficient for inhibition of (target) kinase activity in in vitro experiments. However, we identified a limited number of drug concentration dependent changes of phosphopeptide expression displaying reduced intensity in on- vs pre-treatment samples with increasing tumor concentration in ≥ 3/5 patients (Appendix A). For erlotinib, these included peptides directly related to the target kinase EGFR, as well as PTK2, STAT3 (3/5) and PRKCD (4/5). PRKCD has previously shown to be downregulated by erlotinib [39], and has been, together with STAT3, suggested to sensitize for EGFR inhibition [40]. Confirmation of these peptides in a larger independent dataset would support development of such pharmacogenomic biomarkers. 

We did not focus on a specific tumor type, hypothesizing that drug accumulation and inhibition of on- and off-target kinases may occur independently of primary tumor histology. Based on our finding that drug-specific alterations in peptide phosphorylation are observed in tumor biopsies from patients, while the pre- and on-treatment profiles of individual patients do cluster together, we want to emphasize that inclusion of multiple cancer histologies in this exploratory clinical trial strongly supports the robustness of our finding that PKIs differentially affect tumor phosphoproteomic profiles. In addition, these PKI-specific modifications of the phosphoproteome provide crucial information on the actual biological activity of PKIs in patients as well. 

As a consequence of the study design, response data to the applied (study) treatment was not standardly available; hence, the relation between phosphoproteomic data and clinical response could not be determined for most patients. However, in a patient with head and neck squamous cell cancer, inhibition of two EGFR activation loop (pY869) peptide variants upon erlotinib treatment was observed, as well as inhibition of sites pY1172 and pY1110 located in the C-terminal tail of the kinase. Both sites are indicative of EGFR activity of the target kinase, and inhibition is consistent with the intended target profile of erlotinib. In addition, for three patients with advanced hepatocellular carcinoma whom received the study treatment with sorafenib as part of standard treatment for their disease, inhibition of target kinases related to treatment response in 1 patient, based on reduced activity of known and predicted substrates of MAP2K, EGFR and JAK2 (Appendix A) was observed, while activity persisted in two patients with progressive disease. Evidently, additional prospective studies with larger sample size and available RECIST response data are needed to draw conclusions about the relation between phosphoproteomic profile changes and clinical response to protein kinase inhibitors. Another issue to be addressed for the application of (p-Tyr-)phosphoproteomics for personalized medicine purposes is further development of algorithms to infer kinase or pathway activity. Concerning this matter, kinase-substrate analysis (KSEA) [41], integrated personalized signatures (pCHIPS) [42], kinase activity ranking using phosphoproteomics data (KARP) [43] and integrative inferred kinase activity (INKA) analysis [44] have recently been proposed for kinase pathway activity or drug prioritization, the latter two being applicable to data from a single-shot LC-MS/MS run and INKA being superior for kinase activity ranking. 

## 4. Methods

### 4.1. Study Design 

This single-center molecular profiling study was performed with Institutional Review Board approval in the Amsterdam UMC, location VUmc, Amsterdam, the Netherlands (ClinicalTrials.gov identifier 01636908), in accordance with the Declaration of Helsinki and with the principles of the International Conference on Harmonisation Guidelines for Good Clinical Practice. Written informed consent was obtained from all patients. Patients underwent a tumor needle biopsy before and after 10–14 days of study treatment with 1 of the following PKIs at the approved daily dose: erlotinib (150 mg once daily (qd)), sorafenib (400 mg twice daily (bid)), dasatinib (100 mg qd), vemurafenib (960 mg bid), sunitinib (50 mg qd) or everolimus (10 mg qd) (Figure 1). These PKIs were chosen based on their differential target inhibition and manageable toxicity profile within two weeks of treatment. Study treatment duration of 10–14 days was based on the average minimal time to steady state plasma concentration described in literature [45]. Patients received PKI study treatment either on-label, such as with sorafenib in patients with hepatocellular carcinoma, or off-label prior to initiation of standard systemic (chemo)therapy for their advanced disease, e.g., with dasatinib in a patient with colorectal cancer. In the first case, tumor response was assessed by Response Evaluation Criteria in Solid Tumors (RECIST) version 1.1 [46]. Safety evaluations were performed in all patients with grading of adverse events according to the National Cancer Institute Common Terminology Criteria for Adverse Event, version 4.0 (https://evs.nci.nih.gov/ftp1/CTCAE/). No adverse events CTC grade 3–4 resulted from the biopsies or from study treatment. 

Based on literature reporting sunitinib [27] and erlotinib concentrations [29] in tumors, we aimed to administer each of the drugs under study to five patients to obtain statistically reliable estimates of PKI concentrations in tumors. Ethylenediamine tetraacetic acid (EDTA) plasma and serum were collected for correlative analyses. Optional skin biopsy was taken after 10–14 days of treatment. The primary objective of the study was to determine tumor PKI concentrations after 2 weeks and to correlate these to plasma concentrations. Secondary objectives were to determine the feasibility of MS-based pTyr-phosphoproteomic profiling in pre- and on-treatment tumor biopsies and to evaluate whether drug-specific inhibition of kinases or their substrates could be detected.

Ethics approval and consent to participate: This study was performed with Institutional Review Board approval in Amsterdam UMC, location VUmc, Amsterdam, The Netherlands, in accordance with the Declaration of Helsinki and with the principles of the International Conference on Harmonisation Guidelines for Good Clinical Practice. Written informed consent was obtained from all patients prior to study participation.

### 4.2. Patient Selection and Characteristics 

Adult patients with a histologically confirmed metastatic solid malignancy accessible for biopsy and an indication for standard palliative systemic treatment were eligible. Other inclusion criteria included Eastern Cooperative Oncology Group (ECOG) performance status of 0–2, adequate bone marrow, kidney and liver function. Key exclusion criteria were a history of cardiac disease, uncontrolled hypertension, active infections and wound healing disorders. 

### 4.3. Tumor biopsies

Needle biopsies were taken with up to three passes by an interventional radiologist (supervision by M.R.M.). Non-imaging guided biopsies of well-palpable lesions were taken by an oncologic surgeon (DvdP). Samples were collected by research personnel (ML, HD, MN, JvdM) present on-site allowing snap-freezing of biopsies within 1 minute, followed by storage under −80 °C conditions. 

### 4.4. Drug concentration Measurements

Concentrations of PKIs in tumor, skin, plasma and serum were determined using LC-MS/MS as described elsewhere [27,47]. Data are expressed in µM to allow comparison between tissue and blood concentrations and are based on the conversion of 1 g wet tissue to 1 mL liquid [48]. 

### 4.5. Biopsy Preparation for Phosphoproteomics

Biopsies were cut and processed to tumor lysates for MS-based phopshoproteomics as described elsewhere [13]. Small but representative samples from the same patient could be pooled to obtain sufficient protein for MS-profiling. Tumor tissue lysates were cleared, aliquoted and stored at −80 °C until further use. The bicinchoninic acid assay (BCA) protein assay (ThermoPierce, Rockford, IL, USA) was used to determine protein concentration. Peptide preparation from tumor biopsy lysates, with matched pre- and on-treatment protein input per patient, involved reduction and alkylation of cysteine residues. Hereafter, tryptic digestion, desalting and concentration with subsequent lyophilization of eluted peptides was performed as described elsewhere [13,49,50]. 

### 4.6. Phosphotyrosine Phosphoproteomics Profiling and Data Analysis 

For all biopsy pairs with matched protein input for pre- and on-treatment samples, down-scaled phospho-peptide immunoprecipitation was performed using P-Tyr-1000, an anti-phosphotyrosine antibody coupled to agarose beads (PTMScan, Cell Signaling Technology, Leiden, The Netherlands) as described elsewhere. Peptides were separated by an Ultimate 3000 nanoLC-MS/MS system (Dionex LC-Packings, Amsterdam, The Netherlands) coupled online to a Q Exactive mass spectrometer (Thermo Fisher, Bremen, Germany) [49,50,51]. Control samples of colorectal cancer cell line HCT116 lysate consisting of 1, 5 and 10 mg protein input were added to the 6 drug cohort measurements to monitor MS-performance. MS/MS spectra were searched against the Uniprot human reference proteome FASTA file (release April 2014, 44307 entries, no fragments) using MaxQuant 1.4.1.2. [52]. Enzyme specificity was set to trypsin and up to two missed cleavages were allowed. Cysteine carboxamidomethylation (Cys, +57.021464 Da) was treated as fixed modification and serine, threonine and tyrosine phosphorylation (+79.966330 Da), methionine oxidation (Met, +15.994915 Da), and N-terminal acetylation (N-terminal, +42.010565 Da) as variable modifications. Peptide precursor ions were searched with a maximum mass deviation of 4.5 parts per million (ppm) and fragment ions with a maximum mass deviation of 20 ppm. Peptide, protein and site identifications were filtered at an false-discovery rate (FDR) of 1% using the decoy database strategy. The minimal peptide length was seven amino acids and the minimum Andromeda score for modified peptides was 40, with the corresponding minimum delta score set at 17 [53]. Proteins that could not be differentiated based on MS/MS spectra alone were grouped into protein groups (default MaxQuant settings). (Phospho)peptide identifications were propagated across samples using the match-between-runs option checked. Searches were performed with the label-free quantification option selected. The mass spectrometry proteomics data have been provided to the ProteomeXchange Consortium [54] via the PRIDE partner repository with the dataset identifier PXD008032. A normalization factor derived from the total count of matched protein lysates was applied to scale peptide intensities for each pTyr capture. Cluster analysis of phosphopeptides was performed using hierarchical clustering. Phosphopeptide intensities were normalized to zero mean and unit variance for each phosphopeptide. Subsequently, the Euclidean distance measure was used for phosphopeptide clustering. Normalization of phosphopeptide intensities, reproducibility analyses, and clustering were performed in R. For quantitative reproducibility analyses of replicates, Pearson correlations of log10-transformed normalized intensities of phosphopeptides in the overlap of each comparison were calculated. 

Phosphosite analysis was performed on a subset based on localization probability scores as indicated by MaxQuant. Only class 1 phosphosites (with a localization probability > 0.75) were taken into account for this analysis. Multiple phosphosites were reported in cases where multiple phosphorylations were identified per phosphopeptide. The NetworKIN 3.0 tool [55] was used to attribute phosphopeptide spectral counts, per phosphosite, to a specific kinase [55,56,57]. The filtered and annotated phosphosite table was merged with kinase-substrate relationship data obtained from a NetworKIN-based analysis (predicted relations for phosphosites in the data set). As MaxQuant, for a given phosphopeptide, can predict more phosphosites than the number of actual phosphomodifications of the peptide, the latter number was used to set a maximum multiplication factor for the site-to-count calculation.

## 5. Conclusions

This study constitutes a large-scale evaluation of tumor drug concentrations and phosphoproteomic analysis of serial tumor biopsies from patients with advanced cancer during treatment with PKIs. The tumor PKI concentrations greatly exceeded concentrations required for inhibition of the assumed drug-specific kinase targets. These findings, together with the detected differential regulation of the phosphoproteome per drug implicate that the biological activity of PKIs in patients is due to direct or indirect inhibition of multiple kinases. These findings provide more insight in the clinical mechanism of action of PKIs and indicate that pharmaco-kinetic and dynamic evaluations may further enable future phosphoproteomics-based PKI treatment selection. 

## Figures and Tables

**Figure 1 cancers-12-00330-f001:**
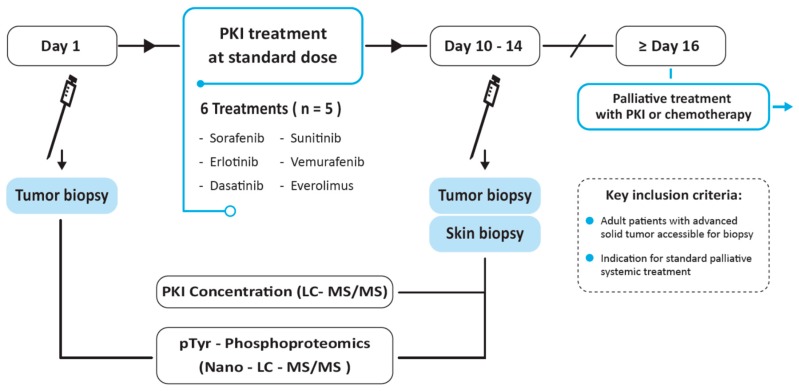
Study design. Patients with advanced solid tumors underwent tumor needle biopsy before and after 10–14 days of treatment with a protein kinase inhibitor, administered as standard treatment for their advanced disease or, off-label, prior to standard palliative chemotherapy. *LC-MS/MS*; liquid chromatography coupled to tandem mass spectrometry.

**Figure 2 cancers-12-00330-f002:**
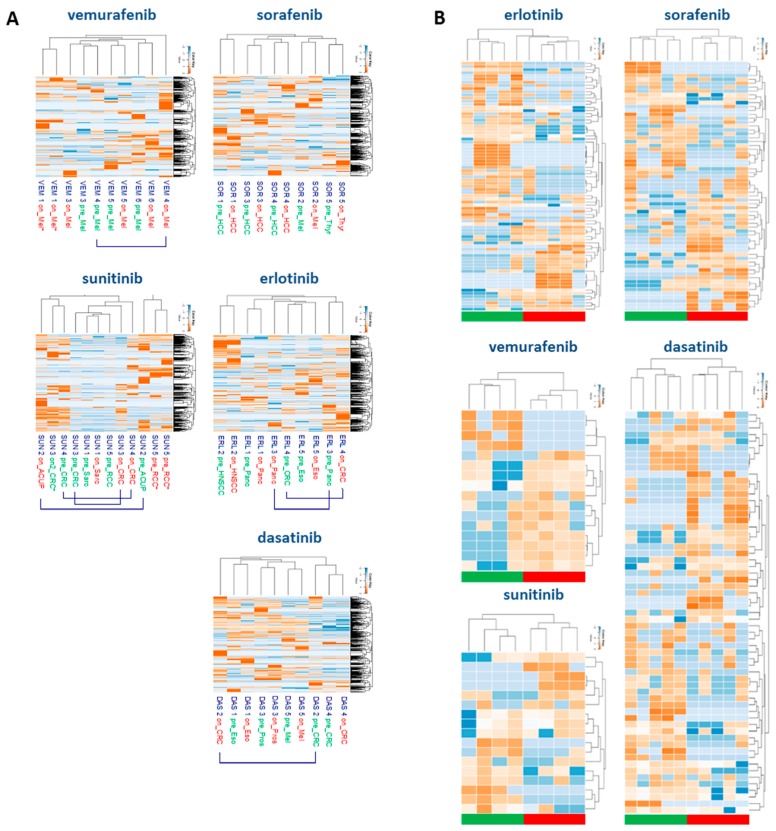
Hierarchical cluster analyses of pTyr-phosphoproteomics data. (**A**) *Unsupervised* hierarchical clustering of the pTyr-phosphoproteome in pre- and on-treatment tumor biopsies. Cluster analysis based on log10-transformed phosphopeptide intensities (red: high abundance, blue: low abundance) shows that samples from individual patients tend to cluster, except for the sunitinib cohort due to limited protein input. Pre- and on-treatment samples are labeled green and red, respectively. *VEM1*, SUN5** denote workflow replicates. (**B**) *Supervised* hierarchical clustering of highly regulated phosphopeptides. Cluster analysis based on up- or downregulated phosphopeptides with a fold-change (intensity on-tx/pre-tx) of > 5 of observed phosphopeptide intensities, in at least 3 patients per cohort. Green blocks indicate pre-treatment samples, red blocks on-treatment samples. Separation of pre- and on-treatment groups is shown for all 5 PKI cohorts.

**Figure 3 cancers-12-00330-f003:**
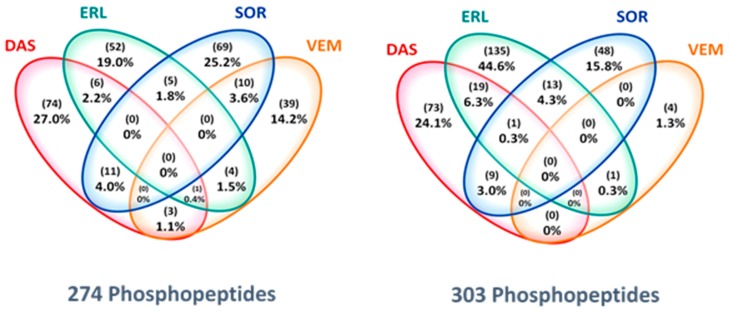
Drug-specificity of PKI-regulated phosphopeptides. Venn diagram depicts the overlap between observed upregulated (**left**) and downregulated (**right**) phosphopeptides per cohort, indicating these are PKI-specific. Analysis based on up- or downregulated phosphopeptides with a fold-change (On-tx/Pre-tx) of > 1.5 of observed phosphopeptide intensities, in at least 3 patients per drug cohort (*VEM*, 4 patients, *SOR/ERL/DAS*, 5 patients; *SUN* not shown).

**Table 1 cancers-12-00330-t001:** Patient characteristics. Clinical characteristics of 31 patients in whom an on-treatment biopsy could be obtained. Patient ID: *SOR*, sorafenib; *ERL*, erlotinib; *DAS*, dasatinib; *VEM*, vemurafenib; *SUN*, sunitinib; *EVE*, everolimus. Gender: *M*, male; *F*, female. Tumor type: *HNSCC*, head and neck squamous cell cancer; *ACUP*, adenocarcinoma of unknown primary origin; (pancreatic) *NET*, neuroendocrine tumor. Radiological response: *SD*, stable disease; *C/P/MR*, complete/partial/mixed response; *PD*, progressive disease; *NE*, not evaluable. pTyr phosphoproteomics could not be performed in 2 VEM patients: for VEM1, no pre-treatment tumor biopsy was taken; for VEM2, on-treatment tumor tissue was insufficient for pTyr profiling. On-treatment tumor drug concentration was determined in these patients.

Patient ID	Gender, Age	Tumor Type	Biopsied Site	Skin Biopsy	Post-Study Treatment (Best Response)
SOR 1	M, 69	Hepatocellular	Liver	+	Sorafenib (SD)
SOR 2	M, 65	Melanoma	Subcutaneous	+	Dacarbazine (PD)
SOR 3	M, 71	Hepatocellular	Liver	+	Sorafenib (PD)
SOR 4	M, 62	Hepatocellular	Liver	+	Sorafenib (PD)
SOR 5	F, 74	Thyroid, papillary	Muscle	+	Sorafenib (SD)
ERL 1	M, 53	Pancreatic	Liver	+	FOLFIRINOX (NE)
ERL 2	M, 57	HNSCC	Subcutaneous	+	Cisplatin/5FU/cetuximab (MR)
ERL 3	M, 71	Pancreatic	Liver	+	FOLFIRINOX (SD)
ERL 4	F, 48	Rectal	Liver	+	CAPOX-B (PR)
ERL 5	M, 68	Esophageal	Liver	+	EOX (PR)
DAS 1	M, 58	Esophageal	Esophagus	+	Gemcitabine/cisplatin (SD)
DAS 2	M, 69	Colorectal	Liver	-	Cetuximab (NE, clinical PD)
DAS 3	M, 62	Prostate	Lymph node	+	Abirateron (SD)
DAS 4	M, 72	Colorectal	Chest wall	+	Irinotecan (PD)
DAS 5	M, 67	Melanoma	Subcutaneous	+	Vemurafenib (PR)
VEM 1	F, 51	Melanoma	Subcutaneous	+	Vemurafenib (SD)
VEM 2	M, 77	Melanoma	Lymph node	+	Vemurafenib (CR)
VEM 3	M, 48	Melanoma	Cutaneous	+	Vemurafenib (NE, clinical PD)
VEM 4	M, 70	Melanoma	Subcutaneous	+	Dabrafenib/trametinib (PR)
VEM 5	M, 61	Melanoma	Subcutaneous	+	Vemurafenib (PR)
VEM 6	M, 82	Melanoma	Lymph node	+	Dabrafenib (PR)
SUN 1	F, 20	Clear cell sarcoma	Lymph node	-	Doxorubicin (PD)
SUN 2	M, 65	ACUP	Subcutaneous	+	Gemcitabin/cisplatin (NE)
SUN 3	M, 59	Colorectal	Subcutaneous	-	CAPOX (NE)
SUN 4	M, 62	Colorectal	Lung	+	CAPOX-B (PR)
SUN 5	M, 69	Renal cell	Lymph node	+	Sunitinib (SD)
EVE 1	M, 67	Renal cell	Subcutaneous	-	Everolimus (SD)
EVE 2	M, 57	Renal cell	Adrenal gland	+	Everolimus/cyclophosphamide (SD)
EVE 3	M, 74	Renal cell	Subcutaneous	+	Sunitinib/dalteparin (PR)
EVE 4	M, 75	Pancreatic NET	Liver	+	Everolimus (SD)
EVE 5	F, 51	NET	Subcutaneous	+	Sandostatin (SD)

**Table 2 cancers-12-00330-t002:** Summary of PKI concentrations after 2 weeks of treatment. PKI concentrations in tumor, skin, plasma and serum per cohort, determined by LC-MS after 10–14 days of treatment. Numbers depict median and range of concentrations based on 5 patients per cohort, unless otherwise indicated. NE, not evaluable. Achieved plasma and serum concentrations at t = 1 week (data not shown) and t = 2 weeks of treatment were highly comparable. Results for individual patients are shown in Appendix A.

PKI Cohort	Tumor µM (Range)	Skin µM (Range)	Plasma µM (Range)	Serum µM (Range)
Sorafenib	10.0 (3.7–22.0)	6.3 (1.4–28.4)	4.8 (3.7–12.1)	6.9 (4.8–17.4)
Erlotinib	4.2 (0.9–10.8)	2.8 (2.1–6.7)	1.2 (0.9–4.0)	1.1 (0.9–4.4)
Dasatinib	2.0 (0.2–64.0)	0.4 (0.2–18.5)*^N^* ^= 4^	0.012 (0.005–0.041)	0.009 (0.017–0.037)*^N^* ^= 3^
Sunitinib	9.0 (2.3–50.0)	4.3 (0.5–9.7)*^N^* ^= 3^	0.1 (0.1–0.2)	0.1 (0.1–0.2)
Vemurafenib	1326 (331–2347)*^N^* ^= 6^	879 (120–2557)*^N^* ^= 8^	98 (65–210)*^N^* ^= 7^	108 (47–242)*^N^* ^= 8^
Everolimus	3.5 (3.4–3.6)*^N^* ^= 2^	NE	NE	NE

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
