# Peer review of "Kinase Inhibitor Treatment of Patients with Advanced Cancer Results in High Tumor Drug Concentrations and in Specific Alterations of the Tumor Phosphoproteome"

_cancers, 2020, doi:10.3390/cancers12020330_

Round 1

Reviewer 1 Report

This paper has some routine charactisation of kinase inhibitors in tissues and some overall conclusions. I think the paper is written well, but the overall significance of this work is over-hyped and must be toned down. The human patient samples are certainly valuable, but the significant of these findings is routine.

The statistics of the groupings are questionable with no P values shown and a change in groups between analysis. Figure 4 should be removed from the paper; it is not robustly supported and with it removed, the paper would flow better. There is over interpretation of the results and this is not helpful to the reader. Section 2.5. should be re-written and cut back to things that are clearly supported.

This sentence Page 8 line 222-224 ‘In this first large-scale MS-based phosphoproteomic analysis of pre- and on treatment tumor needle biopsies obtained in a clinical trial, we studied to what extent these high tumor drug concentrations indeed may alter activity of many more kinases than their known targets’ needs to be toned down. This is not the first time for the type of work and dynamic re-programming of the kinome has been around for some time eg - https://www.ncbi.nlm.nih.gov/pubmed/22500798 (there are many more). This should be mentioned in the introduction.

The overall language used in the discussion should be toned down. The work is interesting and useful, but it needs to be put in context that this work is part of a much bigger puzzle and not a significant leap forward (this is not in itself a bad thing).

minor point - references should be in [] brackets

I support the publication of this work more as a communication with a focus on the key data that is robustly supported.

Author Response

We do thank the reviewers for their supportive feedback and we have adjusted the paper accordingly. Please find our detailed answers below.

Reviewer 1

-We appreciate the comments of the reviewer and do realize that we are excited about our work. We have adjusted the text and especially removed the wording “the first” in the abstract, discussion and conclusion. However, we do not agree with the reviewer that the reported analyses are routine, because data on sequential biopsies in patients during treatment with kinase inhibitors to determine both drug concentrations as well as the effect on the [tyrosine] phosphoproteome have not been published previously. We have clarified this further in the introduction.

-We thank the reviewer on the comment regarding the statistics. 

Because of the limited sample size per cohort and the fact that not all peptides are detected in all patient biopsy pairs (missing values), as we have also found in previous experiments, e.g. ref [21], traditional statistical methods such as the t-test or limma are not reliable. Examining the intensity fold changes is a more robust strategy. In place of p-values, we place constraints on the number of consistent regulation, namely up- and down- regulation in 3 out 5 sample pairs.

As described in section 2.4, fold-changes of phosphopeptide intensities in on-treatment vs. pre-treatment samples were determined for each patient.

To determine differentially up-and down-regulated phosphopeptides between the pre and on-treatment biopsies within each drug cohort, highly regulated peptides were selected by taking only peptides with a] a fold-change of at least 5, as well as b] detected in at least 3 of 5 patients per cohort, into account. The resulting selection of peptides could indeed separate all pre- and on-treatment samples in each drug cohort [Fig 2B].

To analyze whether the up- and down-regulated peptides were different between the cohorts, a less stringent fold-change cut-off of 1.5 was taken, but again only peptides that were regulated in at least 3 of 5 patients per cohort were selected. Despite the less stringent fold-change criterion, the resulting peptides hardly overlapped indicating that the alterations to the phosphoproteome are drug-specific, while based on the use of both selection criteria we deem it highly unlikely that these findings were done by chance.

-As suggested by the reviewer, we removed figure 4 from the paper and added it to the supplements and we adjusted the text of section 2.5.

-We thank the reviewer for the suggestion to add the paper of Duncan et al. and added this reference as nr 14 in the introduction to state the difference in available data on drug-induced alterations in the preclinical versus the clinical setting. Of note, the referenced paper applies chemical proteomics and the dynamic effects on the kinome, rather than the [tyrosine] phosphoproteome, were demonstrated following MEK inhibition in TNBC cell lines, not in tumor tissue from patients.

Reviewer 2 Report

This is an outstanding report of major significance to pharmacological sciences and oncology. The authors provide a careful statement of limitations in approach and outcomes in a clear discussion of key results for the non-specialist reader. The experimental approaches are described, or relevant prior art cited, sufficiently to allow others who are expert in the art to reproduce key aspects of the work. The finding that PKI inhibitor drugs are at concentrations within tumor biopsy sufficient to engage both the assumed primary target as well as known "off-target" kinases raises the question of whether or not clinical efficacy requires a multi-kinase target engagement. A minor suggestion is a more explicit discussion of how the aggregate data might provide insight into adverse event themes observed for the current generation of PKI drugs.

Author Response

We do thank the suggestion of the reviewer and added a sentence in the discussion to use the presented results for new kinase inhibitor drug development to prevent unwanted side effects.

Round 2

Reviewer 1 Report

I accept the authors corrections to the manuscript and think the re-wording is appropriate. A quick read through would be useful, but this can be done at the proof stage.